# Motor Functioning and Intelligence Quotient in Paediatric Survivors of a Fossa Posterior Tumor Following a Multidisciplinary Rehabilitation Program

**DOI:** 10.3390/ijerph19127083

**Published:** 2022-06-09

**Authors:** Mathieu Decock, Robin De Wilde, Ruth Van der Looven, Catharine Vander Linden

**Affiliations:** Physical and Rehabilitation Medicine, Ghent University Hospital, 9000 Ghent, Belgium; ruth.vanderlooven@uzgent.be (R.V.d.L.); catharine.vanderlinden@uzgent.be (C.V.L.)

**Keywords:** fossa posterior tumor, child, rehabilitation, motor functioning, intelligence quotient

## Abstract

Background: Short- and long-term consequences after treatment for childhood fossa posterior tumors are extensively reported in the literature; however, papers highlighting physical function throughout rehabilitation and its correlation with Intelligence Quotient (IQ) are sparse. This study aims to describe the physical functioning and IQ of these survivors, their progression during rehabilitation, and the association with histopathological tumor classification. Additionally, the correlation between gross motor functioning and cognitive functioning was investigated. Methods: This retrospective single-center cohort study included 56 children (35 (62.5%) males and 21 (37.5%) females, with an average age of 6.51 years (SD 4.13)) who followed a multidisciplinary program at the Child Rehabilitation Centre, Ghent University Hospital in the period from 2005 to 2020. Descriptive statistical analysis was performed with the use of non-parametric tests and linear regression to determine the relationship between gross motor functioning and IQ. Results: This report shows impaired motor and intelligence performance in children with a fossa posterior tumor. Although multidisciplinary rehabilitation is beneficial, it is not able to counteract the further decline of several motor skills and intelligence during oncological treatment, more specifically in children with a medulloblastoma. A correlation between gross motor function and total IQ was found. Conclusion: Pediatric survivors of a fossa posterior tumor experience impaired physical and intellectual functions, with more decline during oncological treatment despite simultaneous multidisciplinary rehabilitation.

## 1. Introduction

In the pediatric population, one fourth of cancer diagnoses are brain tumors with an overall incidence between 2.65 and 5.7 per 100,000 children and adolescents [1,2]. More than half of these tumors are located within the fossa posterior, involving the cerebellar hemispheres, brainstem, fourth ventricle or cerebellopontine angle [3]. Various histopathological types of fossa posterior tumors are described according to the World Health Organization (WHO) 2007 classification criteria. The three most common fossa posterior tumors are medulloblastoma (40%), pilocytic astrocytoma (20–35%), and ependymoma (10%) [4]. Treatment of children with fossa posterior tumors has improved significantly during the last 20 years, with an overall current 5-year survival rate above 70% [5,6].

Depending on the WHO classification of the tumor and protocol guidelines, total microsurgical removal of the tumor is often the first objective, followed by complementary treatments such as chemotherapy and radiotherapy. Previous papers have reported extensive short- and long-term consequences after treatment for childhood fossa posterior tumors. In the acute postoperative phase, there is often profound axial hypotonia and ataxia [7], which may persist for more than a year post surgery [8]. Furthermore, it has been reported that pediatric survivors of a fossa posterior tumor show decreased subscale scores on the Bruininks–Osteretsky Test of Motor Performance, Second Edition (BOT-2), compared to normative data [9]. Not surprisingly, impaired physical functioning is most remarkable in the subscale “balance” [9]. A permanent impairment of the postural function, attributed to both the primary tumor and its oncological treatment was demonstrated by Dreneva et al. [10]. Research specifically enlightening the impact of ataxia and the other consequences on physical functioning are rather sparse [11,12]. Nevertheless, physical functioning is important as it may impact the child’s further development, activities of daily living, the ability to return to school and to socialize with his/her friends.

With regard to the neuropsychological outcome, short- and long-term neurocognitive dysfunctions, including memory impairment and IQ decline, have been reported [11]. Risk factors for neurocognitive impairment include the tumor itself, hydrocephalus, age at diagnosis, the extent of tumor surgery, cranial radiation therapy and chemotherapy [11,13].

As therapeutic interventions targeting motor skills and cognitive functions can be effective in this population, children with a history of a fossa posterior tumor usually start with multidisciplinary rehabilitation in specialized centers as soon as possible [14]. Little is known about their physical performance during their rehabilitation period. Forsyth et al. reported a strong relationship between the amount of active practice and gross motor recovery trajectories in children with an acquired brain injury [15]. Yet, understanding the difference rehabilitation can make in the physical outcome of these fossa posterior tumor survivors is a big challenge due to the interaction of individualized rehabilitation programs and protracted oncological treatment. 

To better understand physical and cognitive outcomes in pediatric fossa posterior tumor survivors, this study aimed to investigate:(1)Physical functioning and the intelligence quotient (IQ) using a standardized measure and compared to normative data.(2)The progression of these aforementioned outcome values during a multidisciplinary rehabilitation program.(3)A possible difference in physical functioning and the intelligence quotient (IQ) between the different histopathological tumors (medulloblastoma, ependymoma, astrocytoma).(4)Because cognitive function is related to motor function in small children and children with cerebral palsy, our additional goal is to search for a correlation between motor functioning and cognitive functioning in children with a fossa posterior tumor [16,17].


The null hypotheses of these goals are:(1)There is no difference in the outcomes of pediatric survivors of fossa posterior tumor and normative data.(2)There is no difference in outcomes when comparing the values at the beginning and at the end of the rehabilitation.(3)The histopathological tumor has no effect on physical functioning and IQ.(4)Motor functioning is not related to cognitive functioning.

## 2. Materials and Methods

### 2.1. Participants

A retrospective cohort study design was used, with data collected from the electronic patient record platform of the Children’s Rehabilitation Centre, Ghent University Hospital, Belgium. Children between the age of 0–15 years, diagnosed with a medulloblastoma, ependymoma or pilocytic astrocytoma in the fossa posterior, were selected in the period from 2005 to 2020. They were included in the study if they underwent a (partial) neurosurgical tumor resection followed by a multidisciplinary rehabilitation program (Figure 1). Exclusion criteria were other histopathological types of fossa posterior tumors. This study was approved by the Ghent University Hospital Research Ethics Committee (BC-09217). 

### 2.2. Physical Functioning

To determine physical functioning at the onset and the end of the rehabilitation period, registered data were collected describing muscle strength and range of motion. Muscle strength was graded according to the Medical Research Council Scale. Range of motion was measured using a goniometer based on the neutral zero method. Furthermore, results of three norm-referenced tools to measure gross and fine motor skills in children were assembled: the Bruininks–Oseretsky Test of Motor Proficiency Second Edition (BOT-2), the Peabody Developmental Motor Scale (PDMS-2) and the Purdue Pegboard Test (PPT).

The Bruininks–Oseretsky Test of Motor Proficiency Second Edition (BOT-2) delivers a comprehensive measure of gross and fine motor skills in children aged 4 to 21. The BOT-2 has 53 items organized into four composites: a fine motor manual control composite, manual coordination composite, body coordination composite and strength and agility composite. These composites are further divided into eight subscales [18,19]. 

The Peabody Developmental Motor Scale (PDMS-2) assesses the gross and fine motor skills of young children from birth through 5 years. This test is composed of six subsets, which include reflexes, stationary, locomotion, object manipulation, grasping and visual motor-integration. The PDMS-2 categorizes performance into 1 of 7 categories, with higher scores reflective of better performance [20,21].

The Purdue Pegboard Test (PPT) is an assessment of fingertip dexterity and gross movement of the arm, hand and fingers. The child is seated at a table with the testing board in front of him/her. The testing board consists of two vertical rows of 25 small holes down the center and 4 cups across the top with pins, washers and collars. The clinician administers the following subtests: place as many pins as possible on the row within 30 s with the right hand, with the left hand and with both hands. The last subtest is the use of both hands simultaneously while assembling as many pins, washers and collars as possible within 60 s.

### 2.3. Cognitive Functioning

The Wechsler Intelligence Scale for Children (WISC) is a widely-used standardized tool to assess intelligence in children between the age of 6 and 17 years [22]. Different editions of the test were used in the study time frame (2005–2020), namely, the third and fifth edition. 

The Wechsler Preschool and Primary Scale of Intelligence (WPPSI)-III-NL is an intelligence test for children from the age of 2 year 6 months to 7 years 11 months [23].

The assessment of intelligence in children with problems in the area of verbal communication (e.g., cerebellar mutism, verbal apraxia, foreign language) was performed with the Snijders-Oomen Nonverbal Intelligence test, Revised (SON-R). For the administration of this test, neither the examiner nor the child is required to speak or write. The SON-R 2½-7 is suitable for children between the age of 2, 5 and 7 years, while the SON-R 6-40 is used in older children [24]. 

### 2.4. Statistical Analysis

Descriptive statistical analysis was performed with SPSS version 27.0 (IBM, Armonk, NY, USA) and Excel (Microsoft Office) was used for the graphic illustrations. The comparison of the continuous variables between groups was performed using the non-parametric Wilcoxon (2 groups) or Kruskall–Wallis (more than two groups). The comparison of categorical variables was performed with the non-parametric chi square test. 

Corrected *p*-values lower than 0.05 were considered significant. 

The relationship between physical functioning and cognitive status was assessed by linear regression (Pearson correlation).

## 3. Results

### 3.1. Participants

Fifty-six children were included in our study in the time period of 2005–2020. Thirty-five (62.5%) males and 21 (37.5%) females, with an average age of 6.51 years (SD 4.13) when diagnosed. The average period of the multidisciplinary rehabilitation was 444 days, equaling 1 year and 2 months. Multidisciplinary rehabilitation was started immediately after the neurosurgical procedure and consisted of physiotherapy, occupational, speech-language and neurocognitive training, with a frequency of 3 to 5 times a week (2.5 h/day). The most common tumor histology was a medulloblastoma (50%), followed by a pilocytic astrocytoma (37.5%) and an ependymoma (12.5%).

At presentation in the hospital, nearly 70% of patients needed an emergency surgery with the establishment of an external ventricular drain (EVD) for intracranial hypertension. 

Of the included children, 41.1% had a relapse tumor with a significant predilection for an ependymoma (*p* = 0.028), 19.6% died, most of them diagnosed with a medulloblastoma (10/11 children), and only one child (1/11) died from an ependymoma. 

The characteristics of the population are available in Table 1.

### 3.2. Physical Functioning

To determine physical functioning in our study population with a fossa posterior tumor at the onset and at the end of the rehabilitation period, registered data were collected describing the “balance” and “range of motion”. Furthermore, the results of the BOT-2, PDMS-2 and PPT were assembled and analyzed.

#### 3.2.1. Balance and Range of Motion

Based on the professional experience of the physiotherapist, performance in regard to balance and range of motion was defined as “normal” and “abnormal”. 

Disbalance was mainly associated with medulloblastoma at the onset of rehabilitation (*p* < 0.001) as well as at the end of the rehabilitation period (*p* < 0.076). The range of motion in the upper limbs was seldom disturbed, however, a reduction in the range of motion in the lower limbs was often seen in children with a medulloblastoma. Moreover, this reduction in mobility in the lower limbs was unexpectedly more explicit at the end of the rehabilitation period compared to the beginning of the rehabilitation. The statistical significance (*p*) was calculated between this assessment and the different tumor histology (Table 2).

#### 3.2.2. Gross Motor Functioning

Gross motor functioning was assessed using the Bruininks–Oseretsky Test of Motor Proficiency Second Edition (BOT-2) and the Peabody Developmental Scale (PDMS-2). 

##### The Bruininks–Oseretsky Test of Motor Proficiency Second Edition (BOT-2)

The motor-area composite score distributions for “Body Coordination” and “Strength and Agility” were illustrated using boxplots and scatterplots (Figure 2). The norm-referenced average for each composite is a score of 50.

The composite scale, “Body Coordination” shows a positive tendency throughout the rehabilitation with a mean of 33 (SD 4.947/*n* = 18) at the start versus a mean of 37.8 (SD 8.189/*n* = 10) at the end.

The composite score of “Strength and Agility” did not seem to improve during the rehabilitation with a starting mean of 40.53 (SD 9.716/*n* = 16) and a final mean of 39.91 (SD = 11.149/*n* = 11). The heterogeneity of the data, together with missing data at the end of the rehab, did not allow statistical analysis of pre- and post-intervention.

##### The Peabody Developmental Motor Scale (PDMS-2)

The subscales “Locomotion”, “Object Manipulation” and “Stationary” were used to determine gross motor function in our study cohort of children with a fossa posterior tumor. The norm-referenced average of every subscale is a score between 8 and 12.

The mean of the subscale “Locomotion” in our study population was 4.67 (SD = 2.5/*n* = 9) at the beginning of the rehabilitation and 6.20 (SD = 2.864/*n* = 5) at discharge. Regarding the subscale “Object Manipulation”, a positive tendency throughout the rehabilitation was found with a starting mean score of 8 (SD = 2.366/*n* = 6) and at the end 13 (SD = 4.243/*n* = 2). Finally, the subscale “Stationary” showed no progress with similar scores at the start and the end of the rehabilitation, that is, 7.14 (SD = 2.410/*n* = 7) and 6.50 (SD = 3.536/*n* = 2), respectively (Figure 3).

#### 3.2.3. Fine Motor Functioning

Fine motor functioning was assessed by the BOT-2, the PDMS-2 and the Purdue Pegboard. 

##### The Bruininks–Oseretsky Test of Motor Proficiency Second Edition (BOT-2)

The fine motor-area composite score distributions for “Fine Manual Control” and “Manual Coordination” were illustrated using a scatterplot (Figure 4). The norm-referenced average for each composite is a score of 50.

The mean standard score of “Fine Manual Control” at the beginning was 41.286 (SD 7.544/*n* = 7), which is under the average value of 50. Throughout the rehabilitation there was an amelioration in the fine manual control, with an increase in the mean to 51.25 (SD 6.551/*n* = 4). 

Additionally, the mean standard score of the subscale, “Manual Coordination” is located under the average value at the beginning of the rehabilitation, with a score of 37,857 (SD = 9.720/*n* = 7).

Since there was no available data for post rehabilitation “Manual Coordination”, a comparison could not be made (Figure 4).

##### The Peabody Developmental Motor Scale (PDMS-2)

In the PDMS-2, we focused on the fine motor subtests, “Grasping” and “Visual-Motor Integration”. The norm-referenced average of every subscale is a score between 8 and 12.

The mean score of the subscale “Grasping” in our study was 9 (SD = 3/*n* = 7) at the beginning of the rehabilitation with a mean of 8 (SD = 4.082/*n* = 11) at the end. Although both mean scores lie within the average range, we noticed a small decrease between before and after the rehabilitation. 

The mean of the subscale “Visual-Motor Integration” was 8.91 (SD = 2.3/*n* = 11) at the beginning of the rehabilitation with a substantial decrease after the rehabilitation to 6.20 (SD = 3.347/*n* = 5), below the average range (Figure 5).

##### Purdue Pegboard Test (PPT)

The Purdue Pegboard scores were divided into percentile ranks to provide an insight into the starting values of our group of patients and to compare with the scores at the end of rehabilitation (Figure 6).

Globally, the percentile values of the left and right hand separately, of both hands and the assembly score are most commonly represented in the lowest percentile (0–10). 

After the end of the rehabilitation, there was no improvement in the percentile values with scores over the 50-percentile mark varying from only 10% of the children (scoring of the left hand) to a maximum of 18.2% (scoring of the right hand). 

The mean value scores of the percentile ranks’ score before the rehabilitation were below the average marker of 50. The mean score of the Left Hand was 31.667 (SD 21.602/*n* = 15) and for the Right Hand, the mean score was 27.857 (SD = 27.012/*n* = 14). Scoring for assembly and for Both Hands had a mean value of 35.0 (SD 29.439/*n* = 13) and 21.923 (SD 22.130/*n* = 13), respectively. The mean values of the ranks decreased at discharge from rehabilitation, with a mean value of 18.0 for the Right Hand (SD = 25.832/*n* = 11) and 19.546 (SD = 17.670/*n* = 10) for the Left Hand. The score for assembly had a mean value of 26.429 (SD = 22.678/*n*= 7) and finally, the mean score for the subset of Both Hands was 20 (SD = 23.452/*n*= 6). 

### 3.3. Cognitive Functioning

The results of the WPSI-III and WISC III/V edition are displayed in three categories, total IQ score, verbal IQ score and performance IQ score (Table 3). Regardless of the histopathological type, children with a fossa posterior tumor in our study cohort, had a mean total IQ of 90.94 (SD 12.84), with a verbal IQ of 95.11 (SD 14.998) and performance IQ of 87.41 (SD 13.162) at the start of rehabilitation. Furthermore, no significant correlation could be found between the total, verbal or performance IQ at the start of the rehabilitation and the underlying tumor histology (*p* = 0.364, *p* = 0.145 and *p* = 0.279). 

When analyzing the IQ scores at the end of the rehabilitation, we see a negative tendency in all values (total, verbal and performance) in children with a medulloblastoma, compared to the initial intelligence performance. 

The total IQ score decreased from 92.31 (SD 12.826) to 82.44 (SD 11.980). The verbal IQ score deteriorated from 96.07 (SD 12.899) to 88.33 (SD 12.510) and the performance IQ score decreased from 87.23 (SD 12.969) to 80.56 (SD 80.56). However, this negative tendency in intelligence performance was not seen in the small group of children with a pilocytic astrocytoma. Unfortunately, there was no IQ data available at the end of the rehabilitation of the children with an ependymoma.

### 3.4. Relationship between Intelligence Quotient (IQ) and Motor Functioning

The relationship between IQ and gross and fine motor functioning before the start of the rehabilitation was investigated by using the Pearson correlation (Table 4 and Table 5). 

There was a strong correlation between the Total IQ and the Body Coordination score (BOT-2) with a positive correlation coefficient of 0.519 (*p* = 0.102/*n* = 11). A similar strong correlation was found between the Verbal IQ and the Body Coordination subset with a positive value of 0.528 (*p* = 0.144/*n* = 9). One relationship was strongly negatively correlated, namely, the verbal IQ and the Object Manipulation subset with a Pearson score of −1.000 (*p* < 0.001/*n* = 2); however, we have to be cautious with this conclusion because of the very small number of children. The other investigated correlations did not show a strong correlation (in a positive or negative way). The correlation between the IQ and fine motor functioning showed varying results, with a strong negative correlation between the Visual-Motor Integration (PDMS-2) and the performance IQ. However, as a result of the very small amount of data, no conclusions can be made.

## 4. Discussion

This retrospective study evaluated motor and neurocognitive functioning in children with a history of a fossa posterior tumor. Furthermore, we aimed to highlight the effects of an intensive personalized rehabilitation program in association with oncological treatment. 

First, we analyzed the physical functioning. In line with previous literature, nearly all of the children suffered from disturbed balance at the time of administration in the Child Rehabilitation Centre, which was mainly found in patients with a medulloblastoma [9,10,25]. This postural problem is most likely caused by the tumor invasion or derived from extensive neurosurgical removal. Despite intensive physiotherapy, the same poor results were seen at the end of the rehabilitation period (mean period of 1 year and 2 months). This finding indicates either a more permanent cerebellar dysfunction or the adverse effect of chemotherapy and craniospinal radiotherapy on balance during rehabilitation. Besides the disturbed balance, other gross motor function difficulties were recognized in these children. A decrease in the active and passive range of motion of the upper and lower limbs was observed at the time of administration in the rehabilitation center. As the children started the rehabilitation program before the application of chemotherapy and radiotherapy, this finding could be related to post-neurosurgery immobility, pain or fatigue. Notwithstanding the positive evolution of mobility in the upper limbs, we noted a further deterioration in mobility in the lower limbs (especially ankle dorsiflexion) in children with a medulloblastoma at the end of the rehabilitation period, which was rather unexpected. The adverse effects of chemotherapy agents such as Vincristine on balance and gastrocsoleus flexibility are well known, but seem to be poorly counteracted by intensive rehabilitation [26,27]. This study appears to be the first to describe these clinical findings.

In contrast to the disappointing results on balance and range of motion, a positive evolution throughout the rehabilitation was seen in the composite score distributions for “Body coordination” and “Strength and agility” of the BOT-2, and the subsets Locomotion and “Stationary” of the PDMS-2. Nonetheless, it is important to note that the participants still performed underneath the average on these assessments at the end of the rehabilitation period. Only the results for the PDMS-2 subset, “Object manipulation” increased to the average level. Impaired gross motor function in child survivors of a fossa posterior tumor is well described in many papers [9,28]. However, with our findings, we may assume that these children remain sensitive to physical activity interventions adapted to their medical status. Moreover, gross motor exercise is safe and does not negatively impact the ability to complete chemotherapy and radiotherapy [24,29]. 

With regard to fine motor functioning, a below average level was found on several assessments, more specifically on the subsets, “Fine Manual Control” and “Manual Coordination” of the BOT-2. However, the mean score for “Fine Manual Control” of the BOT-2 improved at the end of the rehabilitation to an average value. Interestingly, the mean score distribution for “Grasping” and “Visual-Motor Integration” of the PDMS-2 was situated in the norm-referenced average range at the beginning of the rehabilitation, but dropped beneath the average at the end. Furthermore, the overall scores of the Purdue Pegboard Test also showed a negative evolution throughout rehabilitation. Many papers have already described the adverse effect of chemotherapy on fine motor skills in children with acute lymphoblastic leukemia, with a significant negative correlation between age and motor or visuomotor performance [30,31,32,33]. A study in which the Purdue Pegboard test was conducted in children with ALL, showed a significant slowing of fine motor speed and dexterity in the dominant hand, nondominant hand and both hands [34]. Our findings in children with a fossa posterior tumor are consistent with these papers, which emphasizes the detrimental impact of chemotherapy on fine motor skills due to neuropathy. 

Secondly, our study analyzed the cognitive function in children surviving a fossa posterior tumor. The mean intelligence quotient (IQ) of the children in our study cohort was 90.94, with a mean verbal IQ (VIQ) of 95.11 and a mean performance IQ (PIQ) of 87.41 at the time of administration in the child rehabilitation center, which is low average in comparison with an age-appropriate standard population. Notwithstanding the fact that we cannot speak of excessive cognitive impairment in these children at the start of their rehabilitation program, we may presume a negative effect of initial increased intracranial pressure, the involvement of the cerebellum and the extent of neurosurgery [11,35]. A growing number of neuroanatomical and functional neuroimaging studies support the role of the cerebellum in a variety of cognitive processes, such as attention, memory, executive functioning, visuo-spatial regulation, learning, language and behavioral-affective modulation [36]. More specifically, “executive functions” (including working memory) have been related to the lateral cerebellar hemispheres, and “attention” to the neocerebellar areas of the hemispheres and the vermis [25]. Despite the fact that these cognitive effects are still controversial in children with cerebellar lesions, we could argue that a lower PIQ compared to the VIQ might be a consequence of executive dysfunction. Another possible explanation may be that decreased fine motor control in performance tasks may lead to lower scores, resulting in a lower PIQ as reported by Grill et al. [35]. However, when we correlated body coordination with IQ, we obtained the highest correlation coefficients between gross motor control and TIQ and between motor control and VIQ, not with PIQ. Analyzing the correlation between fine motor control and PIQ was not conclusive in our study cohort, due to the small amount of data. Previous studies in children with a fossa posterior tumor tend to show less severe cognitive deficits in astrocytoma survivors compared to medulloblastoma survivors [13]. However, at the time of administration in the child rehabilitation center (post-neurosurgery, no adjuvant therapy had started), there was no significant difference in cognitive performance between the children with various tumor types. This interesting finding emphasizes the prominent role of adjuvant oncological therapy in the long-term neurocognitive outcome of these children. After an intensive period of multidisciplinary rehabilitation (mean of 1 year 2 months), in combination with chemotherapy and radiotherapy, we observed a drop of nearly 10% in all IQ variables (TIQ, VIQ, PIQ) in the cohort children with a medulloblastoma. This decline in overall intellectual ability across time corresponds with multiple follow up studies in these children [11,35,37,38]. This decline in IQ seems to be related to failure to acquire information as expected, and not to a deterioration in existing skills [38,39]. 

In conclusion, our study adds fundamental, but also contradictory information. An intensive multidisciplinary rehabilitation program in the post-acute phase during oncological treatment may only have a limited, certain effect on the neurocognitive and physical outcome in children with a medulloblastoma. Of course, although extremely interesting, a comparative age-matched study with fossa posterior tumor survivors who did not receive rehabilitation is not possible in our hospital due to ethical reasons. Chemotherapy and radiotherapy have a known suppressive effect on brain plasticity, cognitive development and physical performance [34,39,40,41]. Therefore, the primary goal of early rehabilitation is to try to (partially) counteract serious decline in physical and cognitive skills, and to prevent secondary complications [42]. Some papers have already supported the positive effects of specialized multidisciplinary rehabilitation with personalized intervention goals in these children [42,43,44]. Our results of moderate improvement in body coordination, locomotion, strength, agility and fine manual control from below average to low average scores, while receiving chemotherapy, confirm the need for rehabilitation. Furthermore, deteriorating results in regard to balance, range of motion, grasping, visual-motor integration and intellectual capacity throughout rehabilitation, are not an indication that multidisciplinary rehabilitation is not successful. It is important to accentuate that rehabilitation is more than exercise. Children also learn to use alternative strategies to compensate for their cognitive and motor function deficits, which facilitates functional independence and may increase participation in age-appropriate activities [45,46].

The limitations of this study need to be acknowledged. First, this study is a retrospective observational cohort study in which results were compared with normative data from the different assessments used. Since the inclusion time frame is a period of fifteen years, some measures were altered in different editions (for instance the WISC-III and WISC-V), which could interfere with our results. Second, there is a considerable amount of missing data, which made it difficult to obtain significant conclusions when we compared results at the beginning and at the end of the rehabilitation period. Therefore, our findings need to be interpreted with caution. A prospective longitudinal study in children with a fossa posterior tumor following rehabilitation would be more ideal to gain complete data sets.

## 5. Conclusions

This report shows the impaired motor and intelligence performance in children with a fossa posterior tumor, and confirms the long-term detrimental effect of oncological treatment. Furthermore, our study indicates that multidisciplinary rehabilitation is beneficial, but is not able to counteract further decline in several motor skills and intelligence in the post-acute phase of treatment, more specifically, in children with a medulloblastoma. Although the efficacy of rehabilitation is moderate, we argue that rehabilitation in the post-acute phase of oncological treatment may increase participation in age-appropriate activity requirements. Longitudinal follow-up studies are warranted to assess the physical and neuropsychological outcomes in this population.

## Figures and Tables

**Figure 1 ijerph-19-07083-f001:**
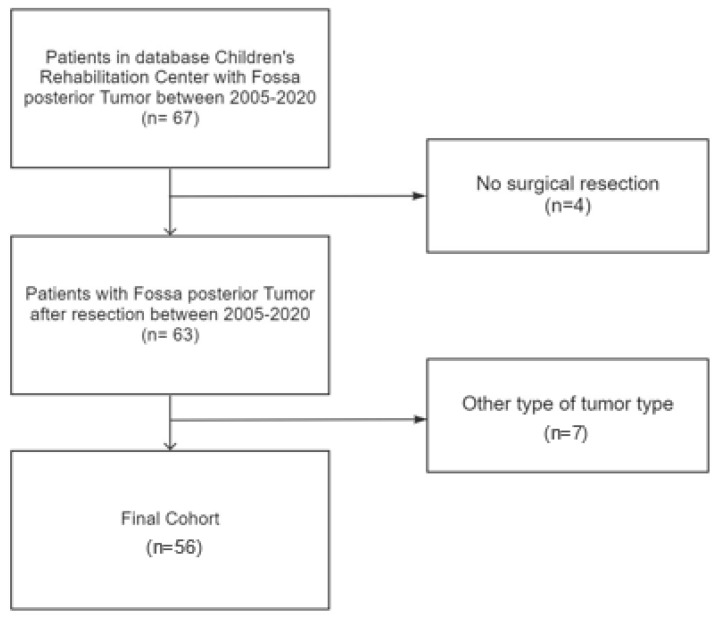
Flowchart of data collection with exclusion criteria.

**Figure 2 ijerph-19-07083-f002:**
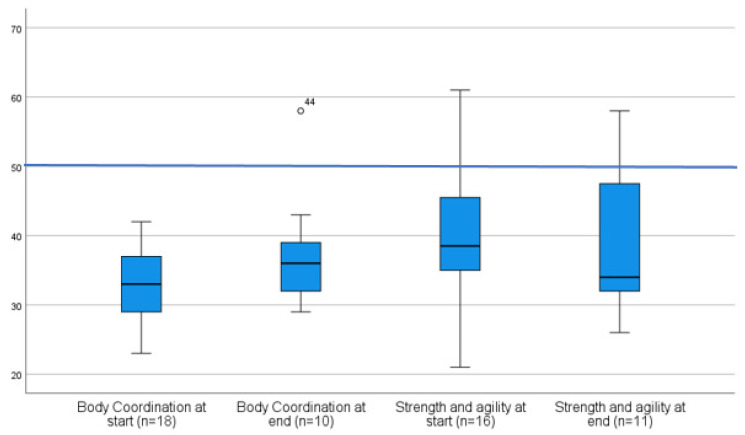
Boxplot of the gross motor-area composite score distributions for “Body Coordination” and “Strength and Agility” of the BOT-2. The horizontal line illustrates the norm-referenced standard score.

**Figure 3 ijerph-19-07083-f003:**
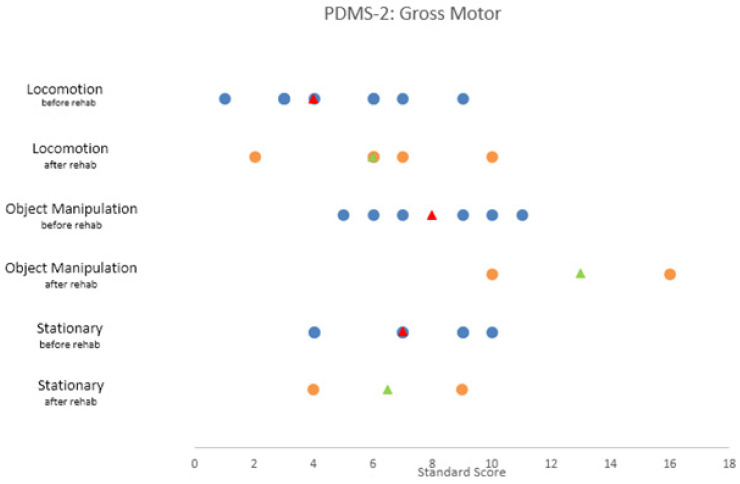
Scatterplot of the gross motor subscale “Locomotion”, “Object Manipulation” and “Stationary” of the PDMS-2. This figure illustrates the values of the scale score at the start and end of the rehabilitation. The median score is illustrated by the triangle symbol. The average score of these subsets is between 8 and 12, and the maximum scale score is 20.

**Figure 4 ijerph-19-07083-f004:**
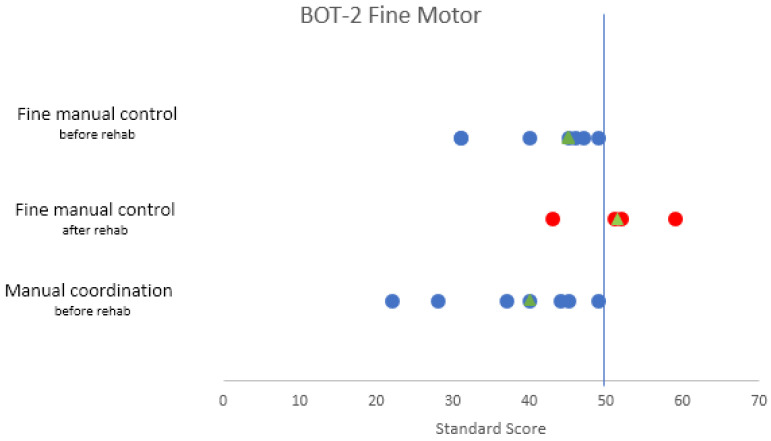
Scatterplot of the fine motor subsets “Fine Manual Control” and “Manual Coordination” of the BOT-2. It illustrates the values of the scale score at the start and end of the rehabilitation. The median score is illustrated by the triangle symbol. The norm-referenced average standard score is 50, illustrated by the vertical line.

**Figure 5 ijerph-19-07083-f005:**
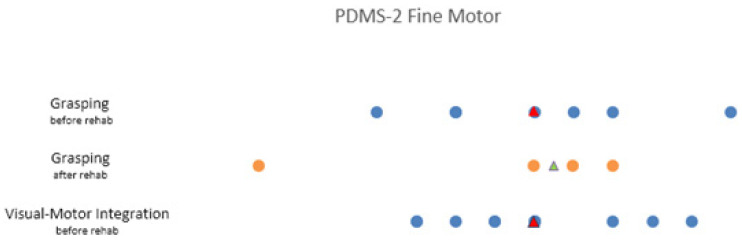
Scatterplot of the fine motor subsets “Grasping” and “Visual-Motor Integration” of the PDMS-2. It illustrates the values of the scale score at the start and end of the rehabilitation. The median score is illustrated by the triangle symbol. The average score in the norm-referenced population of these subsets is between 8 and 12, the maximum scale score is 20.

**Figure 6 ijerph-19-07083-f006:**
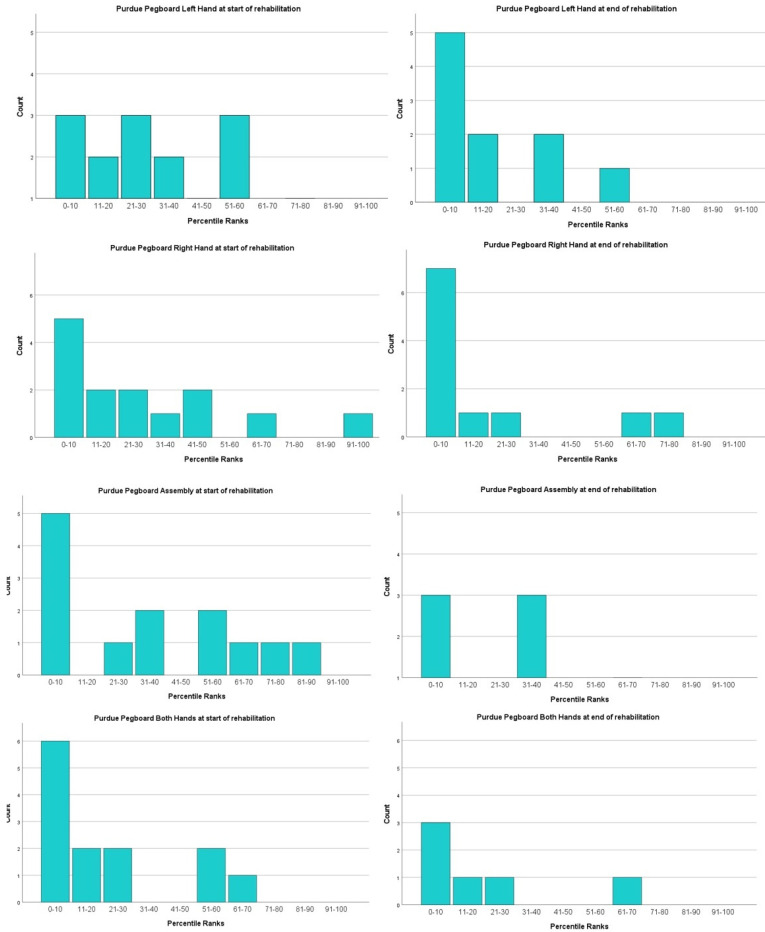
Bar charts of the distribution of the percentile ranks of the PPT. The charts representing different subcategories are illustrated in the rows (Left, Right, Assembly and Both). The left column represents the values before the rehabilitation and the right column illustrates the values after the rehabilitation.

**Table 1 ijerph-19-07083-t001:** Patients’ demographics.

		Number	*%*
Tumor Histology	Ependymoma	7	12.5
	Medulloblastoma	28	50
	Pilocytic Astrocytoma	21	37.5
Sex	Male	35	62.5
	Female	21	37.5
Mean Age (SD)		6.512	4.129
External Ventricular Drain at presentation	Yes	38	67.9
	No	18	32.1
Relapse Tumor	Yes	23	41.1
	No	33	58.9
Deceased	Yes	11	19.6
	No	45	80.4

**Table 2 ijerph-19-07083-t002:** “Balance and Mobility” in our cohort children with a fossa posterior tumor. * No statistical significance could be calculated since only normal values were withheld.

Physical Functioning Parameters	Tumor Histology						Statistical Significance (*p*)
	Medulloblastoma		Ependymoma		Pilocytic Astrocytoma		
	Normal	Abnormal	Normal	Abnormal	Normal	Abnormal	
Balance before rehab (*n* = 16)	0%	68.8% (11/16)	6.3% (1/16)	0%	0%	25% (4/16)	<0.001
Balance after rehab (*n* = 8)	0%	50% (4/8)	12.5% (1/8)	0%	25% (2/8)	12.5% (1/8)	<0.076
Mobility upper limbs before rehab (*n* = 47)	44.7% (21/47)	2.1% (1/47)	10.6% (5/47)	0%	38.3% (18/47)	4.3% (2/47)	<0.637
Mobility upper limbs after rehab (*n* = 27)	55.6% (15/27)	0%	7.4% (2/27)	0%	37% (10/27)	0%	*
Mobility lower limbs before rehab (*n* = 48)	35.4% (17/48)	10.4% (5/48)	10.4% (5/48)	0%	41.7% (20/48)	2.1% (1/48)	<0.138
Mobility lower limbs after rehab (*n* = 28)	35.7% (10/28)	21.4% (6/28)	7.1% (2/28)	0%	35.7% (10/28)	0%	<0.057

**Table 3 ijerph-19-07083-t003:** The three categories of the intelligence quotient (IQ) testing (total, verbal and performance) were categorized by the underlying tumor histology. The mean value of the first test before the rehabilitation and the final test at the end of the rehabilitation is shown.

IQ Score by Tumor Histology							
Tumor Histology		Total IQ before Rehab	Total IQ after Rehab	Verbal IQ before Rehab	Verbal IQ after Rehab	Performance IQ before Rehab	Performance IQ after Rehab
Ependymoma	Mean	90.33		92.67		90.75	
	N	3		3		4	
	Std. Deviation	14.295		12.342		14.431	
Medulloblastoma	Mean	92.31	82.44	96.07	88.33	87.23	80.56
	N	16	9	14	9	13	9
	Std. Deviation	12.826	11.980	12.899	12.510	12.969	10.760
Pylocytic Astrocytoma	Mean	89.25	89.00	94.50	74.00	86.30	101.00
	N	12	3	10	1	10	1
	Std. Deviation	13.505	6.245	19.283	.	14.158	.
Total	Mean	90.94	84.08	95.11	86.90	87.41	82.60
	N	31	12	27	10	27	10
	Std. Deviation	12.842	10.967	14.998	12.635	13.162	12.030

**Table 4 ijerph-19-07083-t004:** The Pearson correlation between the total, verbal and performance intelligence quotient and the different gross motor subscales of the BOT-2 and the PDMS-2 is depicted. The significance was calculated, with the number of available data of each combination shown.

		BOT-2 Bodycoordination	BOT-2 Strength and Agility	PDMS-2 Locomotion	PDMS-2 Object Manipulation	PDMS-2 Stationary
Total IQ	Pearson Correlation	0.519	0.317	0.197	−0.346	−0.002
	Sig. (2-tailed)	0.102	0.372	0.751	0.775	0.998
	N	11	10	5	3	5
Verbal IQ	Pearson Correlation	0.528	0.426	0.078	−1.000	−0.235
	Sig. (2-tailed)	0.144	0.293	0.922	.	0.765
	N	9	8	4	2	4
Performance IQ	Pearson Correlation	0.423	−0.099	0.616	−0.218	0.553
	Sig. (2-tailed)	0.257	0.815	0.268	0.860	0.447
	N	9	8	5	3	4

**Table 5 ijerph-19-07083-t005:** The Pearson correlation between the total, verbal and performance intelligence quotient and the different fine motor subscales of the BOT-2 and the PDMS-2 is depicted. The significance was calculated, with the number of available data of each combination shown.

		BOT-2 Fine Manual Control	BOT-2 Manual Coordination	PDMS-2 Grasping	PDMS-2 Visual-Motor
Total IQ	Pearson Correlation	−0.158	−0.099	−1.000	0.064
	Sig. (2-tailed)	0.899	0.937	.	0.959
	N	3	3	2	3
Verbal IQ	Pearson Correlation	−0.785	−0.094	−1.000	1.000
	Sig. (2-tailed)	0.426	0.940	.	.
	N	3	3	2	2
Performance IQ	Pearson Correlation	0.436	0.500	.	−0.904
	Sig. (2-tailed)	0.713	0.667	.	0.281
	N	3	3	2	3

## Data Availability

Data is available and can be obtained by contacting the corresponding authors.

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
