# Peer review of "Motor Functioning and Intelligence Quotient in Paediatric Survivors of a Fossa Posterior Tumor Following a Multidisciplinary Rehabilitation Program"

_ijerph, 2022, doi:10.3390/ijerph19127083_

Round 1

Reviewer 1 Report

Unfortunately, much of this paper is not clearly explained and the contribution to knowledge is unclear. Many assumptions/presumptions are made without justification. Some particular areas of note are:

Research aim is not clear from the abstract.   Intro L 61-65 What precisely does the lit report on, if not physical functioning? More detail needed to appropriately contextualise the study. Furthermore, the next section 66-73 highlights some physical issues - such ataxia, which seems to contradict the info in L 61-65. I am unsure what the gap in the literature is? Clearer justification and linking to the broader body of literature is needed.   L. 84 what is the significance of cognitive function (CF) and relation to gross motor (GM) function in 'small children' and those with cerebral palsy - I don’t understand the link and how this relates to the research aims. If CF is related to GM function, this needed to be explained in the intro/background.   Research aims are vague ‘evaluation progressing during a multiD rehab program? Determine if there is a difference in PF and IQ. There are too many aims and they are not explained in sufficient detail.   What did the multi-D rehab involve? The disciplines are mentioned, but it is unclear what the program involved beyond this. Similarly, did the multi-D program remain the same over the 15 year time period of the evaluation?   19% died - but at what point during the multi-D program? Details needed.   Due to small sample sizes, it might make more sense not compare different tumor types throughout but present as whole group.   There is no clear evience that the multi-d program actually improved children's functioning - the improvements could have simply been due to chance, therefore this study makes little new contribution to knowledge.   There is some potentially useful info about the specific motor impacts on child, however it is unclear what this adds as previous literature was not thoroughly described in the intro.   ‘gross motor exercise is safe and does not negatively impact… ' how is this relevant to the study as it does not link with aims’? Context needed.   L412-420 I don’t understand what these couple of sentences are saying, or why it is appropriate to ‘presume’ negative impacts of ICP. Similarly 'Seems to be related to failure to acquire…' how/why was this assumption made? This is not clearly explained and appears to be based on authors' opinions.   L462 ‘could assume results would be worse without rehab…’ this assumption is not justified and appears to be based on opinion only.

Reviewer 2 Report

The authors have conducted a study on how much motor function and intelligence quotient is recovered in children following rehabilitation after tumor removal. The authors have conducted many tests but controls are missing and that makes data interpretation difficult. I think authors need to get additional control data to be able to make the manuscript better. 

  1. In line 46 authors should mention number of years. In other words, treatment has improved significantly during the last ------ years?
  2. The authors have tested many qualities such as balance, grasping. The data is quite informative. It will be useful to include a sentence or two in each test describing the kind of test they performed. For example, in fine manual control what exactly did the authors make the patients do and how did they improve after rehabilitation? Basically, I would like the see the authors expand on their findings and discuss implications of the results more. 
  3. Like the authors mention, they don't have comparative normal controls. But it will be useful to mention status of patients before surgery and before chemo/radiotherapy. How much motor function did patients have before diagnosis? How did it compare to normal children? How much of that motor function did they regain or recover after rehabilitation? These need to be discussed. 

Reviewer 3 Report

We thank the authors for submitting their valuable work to the journal. The topic of the paper is interesting and contributes significantly to a better understanding of the subject. However, there are some comments I would make in order to improve the paper's scientific accuracy:

  • Please add a Null Hypothesis to the Objectives paragraph
  • Please add number of Ethics committee approval
  • Please expand on suggested clinical implications of your findings
  • Please format the manuscript according to the IJPERH format

We look forward to receving the revised version of your manuscript.

Kind regards

Round 2

Reviewer 1 Report

The authors have sufficiently addressed the feedback.

Reviewer 2 Report

The authors have addressed my concerns and I support publication.